# SARS-CoV-2 exposure, symptoms and seroprevalence in healthcare workers in Sweden

Ann-Sofie Rudberg [1,2,8], Sebastian Havervall[3,8], Anna Månberg [4], August Jernbom Falk [4], Katherina Aguilera[3], Henry Ng [5], Lena Gabrielsson[3], Ann-Christin Salomonsson[3], Leo Hanke [6], Ben Murrell[6], Gerald McInerney [6], Jennie Olofsson[4], Eni Andersson [4], Cecilia Hellström [4], Shaghayegh Bayati[4], Sofia Bergström[4], Elisa Pin[4], Ronald Sjöberg[4], Hanna Tegel[7], My Hedhammar[7], Mia Phillipson[5], Peter Nilsson [4,9], Sophia Hober [7,9] & Charlotte Thålin [3,9✉]

SARS-CoV-2 may pose an occupational health risk to healthcare workers. Here, we report the seroprevalence of SARS-CoV-2 antibodies, self-reported symptoms and occupational exposure to SARS-CoV-2 among healthcare workers at a large acute care hospital in Sweden. The seroprevalence of IgG antibodies against SARS-CoV-2 was 19.1% among the 2149 healthcare workers recruited between April 14th and May 8th 2020, which was higher than the reported regional seroprevalence during the same time period. Symptoms associated with seroprevalence were anosmia (odds ratio (OR) 28.4, 95% CI 20.6–39.5) and ageusia (OR 19.2, 95% CI 14.3–26.1). Seroprevalence was also associated with patient contact (OR 2.9, 95% CI 1.9–4.5) and covid-19 patient contact (OR 3.3, 95% CI 2.2–5.3). These findings imply an occupational risk for SARS-CoV-2 infection among healthcare workers. Continued measures are warranted to assure healthcare workers safety and reduce transmission from healthcare workers to patients and to the community.

[1] Department of Neurology, Danderyd hospital, Stockholm, Sweden. [2] Department of Clinical Neuroscience, Karolinska Institutet, Stockholm, Sweden. [3] Division of Internal Medicine, Department of Clinical Sciences, Karolinska Institutet, Danderyd Hospital, Stockholm, Sweden. [4] Division of Affinity Proteomics, Department of Protein Science, KTH Royal Institute of Technology, SciLifeLab, Stockholm, Sweden. [5] Department of Medical Cell Biology, Uppsala University, SciLifeLab, Uppsala, Sweden. [6] Department of Microbiology, Tumor and Cell Biology, Karolinska Institutet, Stockholm, Sweden. [7] Division of Protein Technology, Department of Protein Science, KTH Royal Institute of Technology, Stockholm, Sweden. [8]These authors contributed equally: Ann-Sofie Rudberg, Sebastian Havervall. [9]These authors jointly supervised this work: Peter Nilsson, Sophia Hober, Charlotte Thålin. ✉email: charlotte.thalin@sll.se

The novel coronavirus coined severe acute respiratory syndrome coronavirus-2 (SARS-CoV-2), causes the coronavirus disease 2019 (COVID-19). The World Health Organization (WHO) declared COVID-19 to be a pandemic on March 11, 2020[1] and currently more than 24 million cases have been reported, leading to over 820,000 deaths. The first laboratory confirmed case of COVID-19 infection in Sweden was observed 31 January 2020, escalating to 22,317 cases and more than 5000 deaths as of 17 June 2020[2]. Epidemiology of the COVID-19 infection is, however, largely based on cases requiring hospitalization, and little is known about the true extent of the disease. Serological population-based investigations provide a useful tool in the estimation of the number of individuals who have been infected with the SARS-CoV-2 virus and who may be at reduced risk for re-infection. Several estimations of population-based seroprevalence are emerging, ranging from 4.4% in France[3], 4.6% in Los Angeles[4], and 7.3% in Stockholm[2], all from April–May 2020.

The Swedish main objective with the actions coupled to the COVID-19 pandemic is similar to that of most other countries; to reduce the spread of the infection. However, instead of a full lock down, the strategy is to keep parts of the society open. Events where more than 50 people take part are banned, but the majority of actions to reduce the spread rely upon voluntary compliance with the Public Health Authority's evolving set of recommendations[2]. These recommendations are non-compulsory but individuals are expected to follow them, despite the lack of fines for any failure. The inhabitants are urged to work from home if possible and to avoid travels. Further, limited social contacts are encouraged, specifically among individuals with the greatest risk for COVID-19 complications. In contrast to many other countries, however, preschools and grade schools remain open.

Healthcare workers (HCWs) are exposed to the virus at a greater extent than society as a whole and may be considered at an elevated risk of infection. During the previous SARS epidemic, HCW comprised more than 20% of all cases[5–7]. This raises concerns about the safety of front-line HCW and the risk of healthcare system collapse as well as transmission from healthcare settings to the community. Little is, however, known about the occupational risk of HCW to SARS-CoV-2 infection, but the few emerging studies report relatively low seroprevalences, ranging from 1.6% to 11.0%[8–14].

Here we report the seroprevalence of SARS-CoV-2 IgG antibodies among HCW at a large acute-care hospital in Sweden. We furthermore assess associations between seroprevalence and self-reported symptoms and occupational exposure to SARS-CoV-2.

## Results

**Characteristics of study participants**. A total of 2149 HCW were included in the study. The majority of study participants were women (85%) and the mean age was 44 (SD 12) years. Patient contact was reported by 1764 participants (85%), and 962 (46%) participants reported COVID-19 patient contact. The group with patient contact comprised 439 (25%) physicians, 636 (36%) nurses, 428 (24%) assisting nurses and 254 (14.5%) other healthcare staff (occupation was missing in 0.5%). (Table 1).

**Seroprevalence of antibodies against SARS-CoV-2**. Overall, 410 study participants (19.1%) were seropositive for IgG. There was no difference in age or sex between seropositive and seronegative individuals (see above).

**Self-reported symptoms**. Interestingly, among seropositive individuals, 37 individuals (9%) reported no symptoms at all, 320 individuals (78%) reported mild symptoms and only 53 individuals

### Table 1 Demographics, symptomatology, and occupational exposure of seropositive and seronegative study participants.

|  | All | Seronegative | Seropositive |
|---|---|---|---|
|  | n = 2149 | n = 1739 | n = 410 |
| Age (years), mean (SD) | 44 (12) | 44 (12) | 43 (12) |
| Age: Missing, n (%) | 2 (0) | 1 (0) | 1 (0) |
| Female, n (%) | 1815 (85) | 1475 (85) | 340 (83) |
| Male, n (%) | 331 (15) | 261 (15) | 70 (17) |
| Sex: Missing, n (%) | 3 (0) | 3 (0) | 0 (0) |
| *Symptoms since 1st January 2020, n (%)* |  |  |  |
| Fever | 538 (25) | 304 (17) | 234 (57) |
| Headache | 991 (46) | 722 (42) | 269 (66) |
| Anosmia | 283 (13) | 66 (4) | 217 (53) |
| Ageusia | 289 (13) | 85 (5) | 204 (50) |
| Cough | 716 (33) | 503 (29) | 213 (52) |
| Malaise | 912 (42) | 644 (37) | 268 (65) |
| Common cold | 738 (34) | 557 (32) | 181 (44) |
| Abdominal symptoms | 382 (18) | 261 (15) | 121 (30) |
| Sore throat | 822 (38) | 660 (38) | 162 (40) |
| Shortness of breath | 303 (14) | 205 (12) | 98 (24) |
| Missing | 0 (0) | 0 (0) | 0 (0) |
| *Degree of symptoms, n (%)* |  |  |  |
| Mild | 1573 (73) | 1253 (72) | 320 (78) |
| Severe | 116 (5) | 63 (4) | 53 (13) |
| No symptoms | 460 (21) | 423 (24) | 37 (9) |
| Missing | 0 (0) | 0 (0) | 0 (0) |
| *Exposure, n (%)* |  |  |  |
| Physicians | 439 (21) | 355 (21) | 84 (21) |
| Nurses | 636 (31) | 497 (30) | 139 (35) |
| Assisting nurses | 428 (21) | 319 (19) | 109 (27) |
| Other healthcare staff | 254 (12) | 220 (13) | 34 (9) |
| Employees with no patient contact | 305 (15) | 279 (17) | 26 (7) |
| Direct patient contact | 1764 (85) | 1393 (83) | 371 (93) |
| Direct contact with COVID-19 patients | 962 (46) | 734 (44) | 228 (57) |
| Missing | 80 (4) | 67 (4) | 13 (3) |

Source data are available as Source Data file.

(13%) reported severe symptoms. The most frequently reported symptoms in this group were headache (66%), malaise (65%), fever (57%), anosmia (53%), cough (52%) and ageusia (50%), while abdominal pain (30%) and dyspnea (24%) were the least reported symptoms (Fig. 1).

Symptoms with the strongest association to seroprevalence were anosmia (OR 28.4; 95% CI 20.6–39.5), ageusia (OR 19.2; 95% CI 14.3–26.1) and fever (OR 6.3; 95% CI 4.9–8.0). The only symptom that did not differ in prevalence between seronegative and seropositive participants was sore throat (OR 1.1; 95% CI 0.9–1.3). Combining the symptoms with the strongest association to seroprevalence into the triad anosmia and/or ageusia, malaise, and fever rendered a high predictive value with an OR of 18.6 (95% CI 12.9–27.2), a PPV of 0.75 and an NPV of 0.86. In light of the recently published large population-based real-time tracking of self-reported symptoms predicting COVID-19[15], which presented a high predictive value of the symptom set anosmia, fatigue, persistent cough and loss of appetite, the symptom triad anosmia and/or ageusia, malaise and cough (loss of appetite was not included in our questionnaire) was also evaluated. This symptom triad rendered a lower, albeit still high, prognostic value with an OR of 11.9 (95% CI 8.4–17.1), a PPV of 0.67 and an NPV of 0.85. The highest predictive value was, however, found when

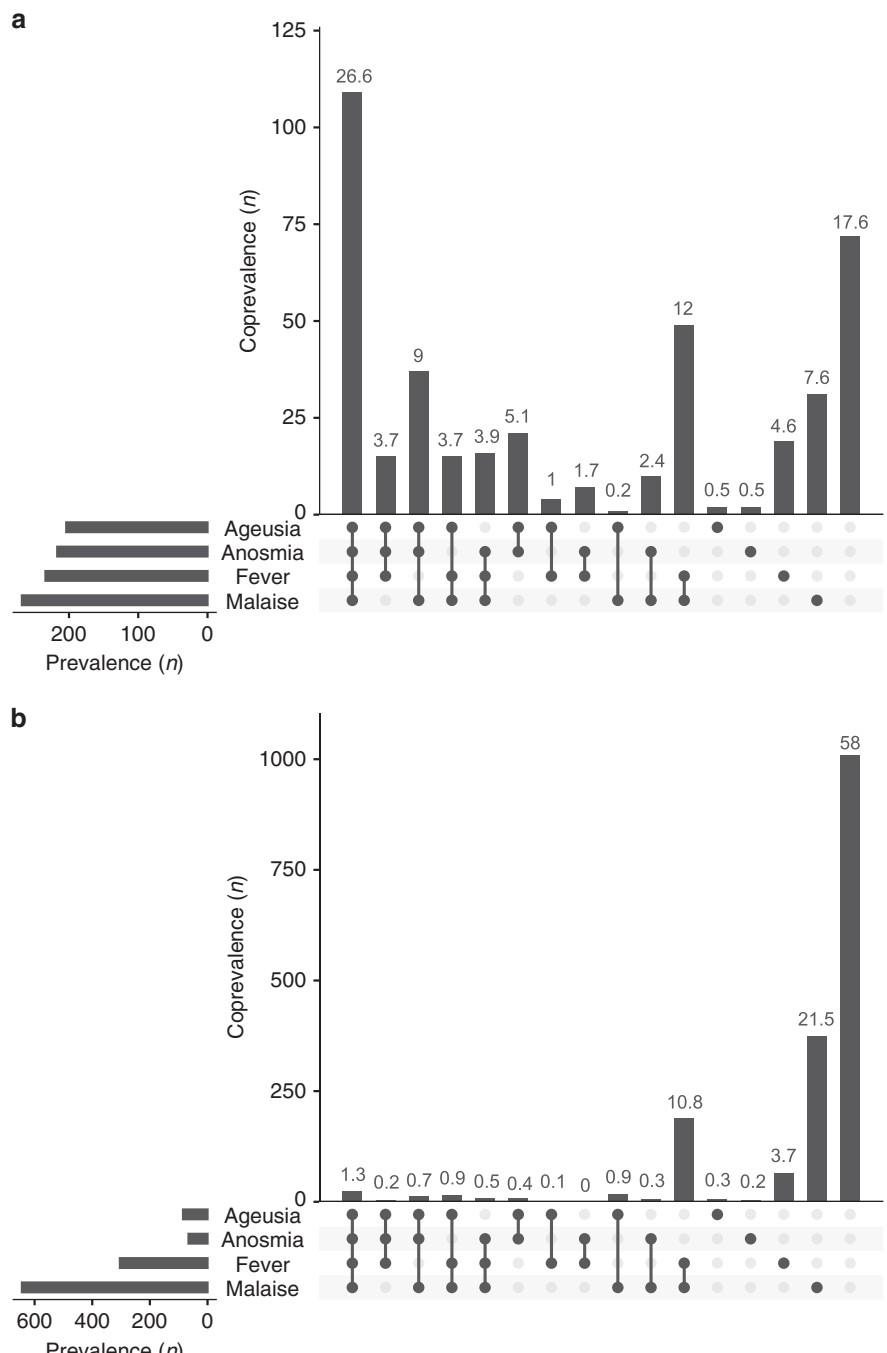

**Fig. 1 Symptomatology.** Symptomatology in seropositive (**a**; *n* = 410) and seronegative (**b**, *n* = 1739) individuals. Horizontal bars to the left represent the total number of participants in each group reporting the specifically denoted symptom. Vertical bars show the total number of participants in each group reporting symptoms symbolized with black dot(s) in the corresponding column. The percentage of participants reporting symptoms symbolized with black dot(s) in the corresponding column is presented above the bars. Source data are available as Source Data file.

combining anosmia and/or ageusia, with an OR of 21.9 (95% CI 16.5–29.3), a PPV of 0.70, and an NPV of 0.905. (Fig. 2).

**Self-reported patient-related work and occupational exposure to COVID-19.** Seroprevalence was associated to patient-related work, with 21% among 1764 study participants with patient contact vs. 9% among 305 study participants without patient contact (OR 2.9, 95% CI 1.9–4.5). Interestingly, this association remained significant regardless of COVID-19 patient contact (OR 3.3, 95% CI 2.2–5.3) or non-COVID-19 patient contact (OR 2.3, 95% CI 1.5–3.8), although the association was significantly

stronger if contact with COVID-19 patients compared to contact with non-COVID-19 patients (OR 1.4, 95% CI 1.1–1.8). (Fig. 3).

The association between patient contact and seroprevalence remained significant regardless of whether the occupation was physician (OR 2.5, 95% CI 1.6–4.2), nurse (OR 3.0, 95% CI 1.9–4.9) or assisting nurse (OR 3.7, 95% CI 2.3–6.0), (Fig. 4), but the seroprevalence was higher among assisting nurses (25%) and nurses (22%) than among physicians (19%) and other medical staff (13%).

The seroprevalence listed by sex, symptom and occupational exposure is shown in Table 2.

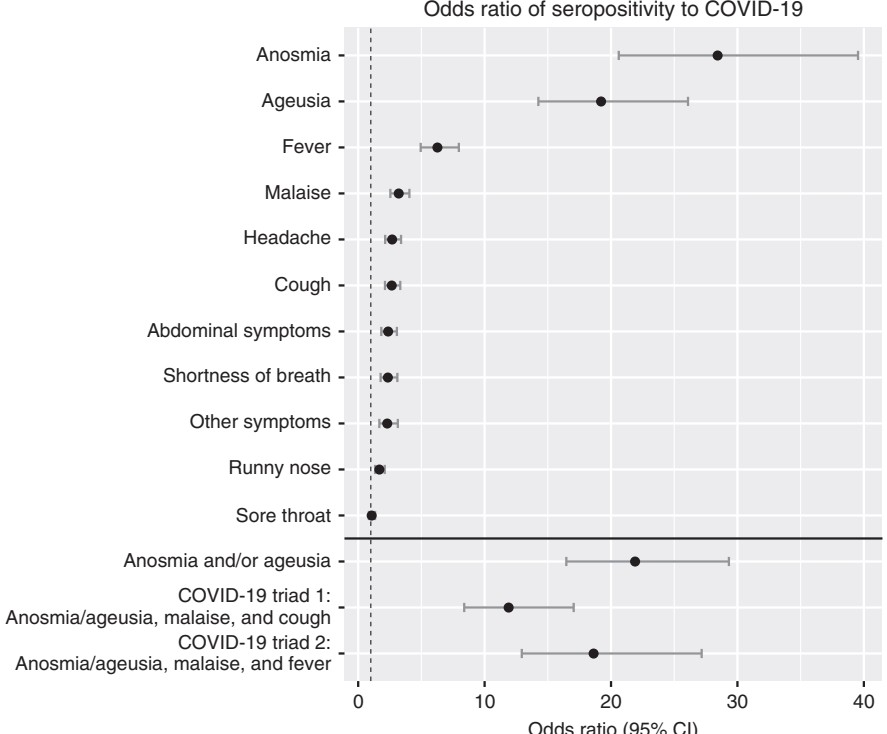

**Fig. 2 Associations between prior symptoms and seroprevalence of SARS-CoV 2 IgG antibodies.** Odds ratios of seropositivity for individually reported symptoms. Odds ratios were calculated using two-sided Fisher's exact test with $n = 2149$ independent individuals. No adjustment for multiple comparisons was applied. Data are presented as odds ratios and 95% confidence intervals. Source data are available as Source Data file.

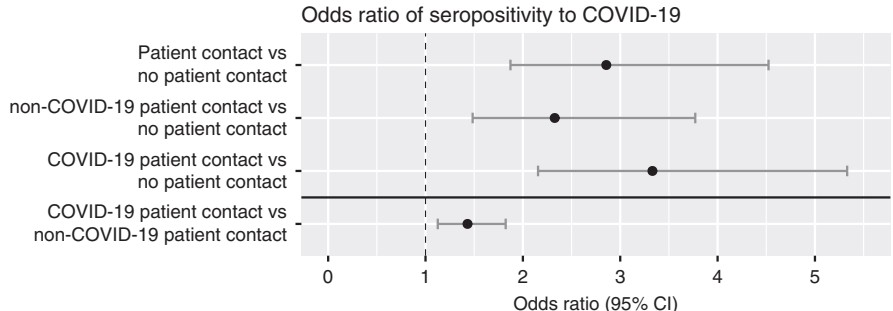

**Fig. 3 Association between occupational exposure and seroprevalence of SARS-CoV 2 IgG antibodies.** Odds ratios of seropositivity given patient contact, non-COVID-19 patient contact, or COVID-19 patient contact compared to no patient contact, and (under the horizontal black line) given COVID-19 patient contact compared to non-COVID-19 patient contact. Odds ratios were calculated using two-sided Fisher's exact test with $n = 2149$ (patient contact vs no patient contact), $n = 1107$ (non-COVID-19 patient contact vs no patient contact), $n = 1267$ (COVID-19 patient contact vs no patient contact), and $n = 1764$ (COVID-19 patient contact vs non-COVID-19 patient contact) independent individuals. No adjustment for multiple comparisons was applied. Data are presented as odds ratios and 95% confidence intervals. Source data are available as Source Data file.

## Discussion

This large serological investigation reports symptoms associated with SARS-CoV-2 infection, which may aid in healthcare personnel screening guidance and in the recommendations of self-isolation. The results furthermore support an occupational risk of SARS-CoV-2 transmission to HCW, exceeding the risk presented in recent studies[8–14]. Among the enrolled 2149 HCW, one in five were seropositive, suggesting prior or still ongoing infection with SARS-CoV-2. The seroprevalence was significantly higher in HCW with patient contact than in those without patient contact, and the seroprevalence among HCW without patient contact was in line with that reported in the general population of Stockholm using the same serological test during the same time period[2]. The seroprevalence was also higher among HCW with COVID-19 patient contact than among HCW with non-COVID-19 patient

contact, implying a patient-related transmission of SARS-CoV-2 to HCW.

The portion of seropositive HCW reporting no prior symptoms is in this cohort lower than several outbreak investigations where 20–50% of COVID-19 infections are suggested to remain subclinical[9,16]. However, the vast majority of HCW reported mild symptoms that are difficult to distinguish from other respiratory infections. Identification of symptoms predictive of SARS-CoV-2 infection, single or in combination, is essential for the quest of screening guidance and in the recommendations of self-isolation to prevent further spread. In the large population-based study including 9282 HCW in the United States, 92% reported having at least one symptom among fever, cough and dyspnea[17], and this symptom triad has been reported hallmark symptoms of COVID-19 infection[18–20]. Although fever was one of the most common

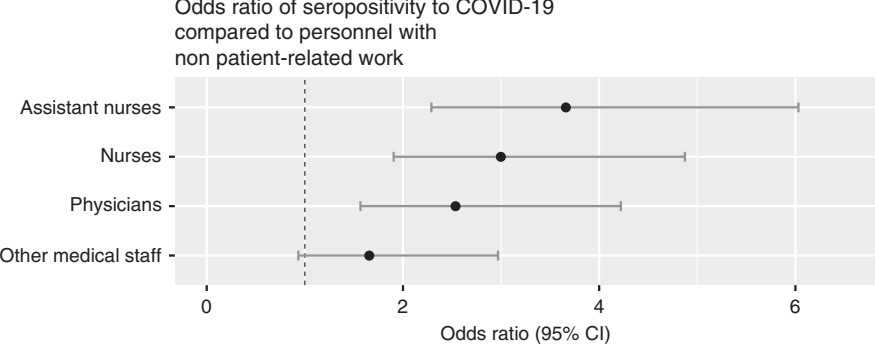

**Fig. 4 Associations between occupations and seroprevalence of SARS-CoV 2 IgG antibodies among HCW with patient contact.** Odds ratios of seropositivity for groups of HCW with patient contact, compared to personnel without patient contact. Odds ratios were calculated using two-sided Fisher's exact test with n = 733 (assistant nurses vs no patient contact), n = 941 (nurses vs no patient contact), n = 744 (physicians vs no patient contact), and n = 559 (other medical staff vs no patient contact) independent plasma samples. No adjustment for multiple comparisons was applied. Data are presented as odds ratios and 95% confidence intervals. Source data are available as Source Data file.

**Table 2 Seroprevalence listed by sex, symptom and occupational exposure.**

|  | n | Seronegative, n (%) | Seropositive, n (%) |
|---|---|---|---|
| **Sex** | | | |
| Female | 1815 | 1475 (81) | 340 (19) |
| Male | 331 | 261 (79) | 70 (21) |
| Missing | 3 | 3 (100) | 0 (0) |
| *Symptoms since 1st January 2020* | | | |
| Fever | 538 | 304 (57) | 234 (43) |
| Headache | 991 | 722 (73) | 269 (27) |
| Anosmia | 283 | 66 (23) | 217 (77) |
| Ageusia | 289 | 85 (29) | 204 (71) |
| Cough | 716 | 503 (70) | 213 (30) |
| Malaise | 912 | 644 (71) | 268 (29) |
| Common cold | 738 | 557 (75) | 181 (25) |
| Abdominal symptoms | 382 | 261 (68) | 121 (32) |
| Sore throat | 822 | 660 (80) | 162 (20) |
| Shortness of breath | 303 | 205 (68) | 98 (32) |
| Missing | 0 | 0 (0) | 0 (0) |
| *Degree of symptoms* | | | |
| Mild | 1573 | 1253 (80) | 320 (20) |
| Severe | 116 | 63 (54) | 53 (46) |
| No symptoms | 460 | 423 (92) | 37 (8) |
| Missing | 0 | 0 (0) | 0 (0) |
| *Exposure* | | | |
| Physicians | 439 | 355 (81) | 84 (19) |
| Nurses | 636 | 497 (78) | 139 (22) |
| Assisting nurses | 428 | 319 (75) | 109 (25) |
| Other healthcare staff | 254 | 220 (87) | 34 (13) |
| Employees with no patient contact | 305 | 279 (91) | 26 (9) |
| Direct patient contact | 1764 | 1393 (79) | 371 (21) |
| Direct contact with COVID-19 patients | 962 | 734 (76) | 228 (24) |
| Missing | 80 | 67 (84) | 13 (16) |

Source data are available as Source Data file.

symptoms among the seropositive HCW in this study, anosmia and ageusia were the symptoms most predictive of SARS-CoV-2 infection. Of these, anosmia is emerging as a key symptom in COVID-19[9,15,21]. The results from the present study are in line with the recent large multinational population-based cohort study investigating potential symptoms of COVID-19, reporting a strong association between anosmia, ageusia and COVID-19[15].

Furthermore a strong association between the symptom combinations anosmia and/or ageusia, malaise and fever as well as anosmia and/or ageusia, malaise and cough and seroprevalence was found, suggesting that these symptoms should be included in routine screening guidance.

The occupational risk of HCW to be infected with SARS is well documented[22–24]. Reports now highlight the risk of occupational transmission of SARS-CoV-2 as well[25–31], and contamination of SARS-CoV-2 RNA has been demonstrated to be widespread across hospital environmental surfaces[32] and air samples from both ICU and general wards[33]. Emerging serological investigations, however, document relatively low seroprevalences among HCW, ranging from 1.6% in Germany[12] and 3.8% in Wuhan[8], the epicenter of the COVID-19 pandemic in China, to 7.6 and 11% in the severely affected countries Belgium[10] and UK[11], which are all significantly lower than the seroprevalence found in this cohort of HCW.

Although the discrepancies between the high seroprevalence in this study compared to prior studies may be explained by different study designs and serological assays, divergencies in infection prevention and control precautions may have contributed. For instance, the hospital investigated in the Belgian study[10], with a study design and sample size similar to this study, implemented a number of prevention measures not undertaken at the hospital in this study. These included RT-PCR testing and subsequent isolation of infected HCW, and RT-PCR testing of all in-hospital patients, regardless of typical COVID-19 symptoms. Notably, Garcia-Basteiro et al.[9] found that 23.1% of seropositive HCW had been asymptomatic, and 9% of seropositive HCW in our cohort had no prior symptoms at all. The lack of RT-PCR testing in HCW and in-house patients without typical COVID-19 symptoms may have contributed to a substantial number of SARS-CoV-2 positive cases among practicing HCW as well as difficulties in distinguishing between COVID-19 and non-COVID-19 patients. Indeed, the seroprevalence among HCW with non-COVID-19 patient contact remained elevated compared to the group of HCW without any patient contact, suggesting transmission in non-COVID-19 wards. The high seroprevalence among assistant nurses and nurses, regardless of work in covid or non-covid wards, further supports transmission from patients to HCW when considering that these occupations involve the most patient-near contact. Moreover, PPE was only worn by HCW in contact with patients with known or suspected COVID-19, and aerosol filtering face masks were restricted to AGP. Notably, emerging data implies that SARS-CoV-2 may spread both via direct contact as well as indirectly through contaminated objects and aerosol[30]. Furthermore, a recent population-based study using a novel mobile-based application to

examine the risk of testing positive for COVID-19, which included over 90 000 HCW[31] found that the highest risk of transmission to HCW was among HCW who reused PPE. Due to shortage of PPE, reuse and sharing of manually cleaned and disinfected face shields was implemented at the hospital in this study. Although the effect of restricted RT-PCR testing and PPE usage was not evaluated, these factors may have contributed to the relatively high seroprevalence among HCW. Importantly, the surge capacity for adequate RT-PCR testing and PPE was hampered nationally by limited testing availability and PPE supply, and was therefore likely similar in other hospitals and healthcare settings around the nation. This emphasizes the urgency in continuous and adequate preparations for large-scale outbreaks and pandemics, including ensuring the availability of diagnostic capacities and PPE.

The results observed in this study are strengthened by the large sample size of individuals representing all departments at the hospital. Furthermore, the response rate of self-reported data was close to 100%. The self-reporting of occupational exposure to SARS-CoV-2 was found to be superior to occupational location in the hospital database since a large portion of study participants are re-located between wards during the COVID-19 pandemic. The results are further strengthened by the use of a highly sensitive and specific laboratory-based antibody assay. This is in contrast with many other serological test methods, where the sample set used for evaluation is very low and thereby also the certainty of the test method[34].

The study also has several limitations worth noting. Seroprevalence is dynamic, and the data presented in this study represents the prevalence of prior or still ongoing infection in April-May 2020. A new cross-sectional investigation will yield different results. Another limitation lies in the nature of self-reported data, yielding a risk of recall bias. Anosmia and ageusia have been widely pointed out as potential COVID-19 symptoms in media, which may have influenced responses. Participants were asked to document symptoms over the prior 3-4 months, and it is not certain that reported symptoms were caused by SARS-CoV-2 infection in seropositive participants, who may have been infected by other respiratory viruses in addition to SARS-CoV-2 in the months prior to study inclusion. The grading of symptoms as "mild" or "severe" were not defined further in the questionnaire, and are therefore subjective and should be interpreted with caution. The options of occupation, patient contact and exposure to COVID-19 patients were objective variables, but a limitation is that no information regarding the type of contact, such as time in the same room and frequency of patient contact is available. Study participation was voluntary, and a selection bias cannot be excluded. Individuals with prior symptoms may have been more likely to participate, and employees not at work due to illness during the study period of three weeks were not able to participate, which could have influenced the overall seroprevalence. However, possible selection bias would apply to all sub groups, and comparisons between groups were likely unaffected. Finally, although a clear association was observed between seroprevalence, patient contact, COVID-19 patient contact and occupation, exposures to COVID-19 positive individuals outside of the hospital were not documented. A possible cluster spread among study participants regardless of occupational exposure to SARS-CoV-2 furthermore cannot be ruled out.

In conclusion, anosmia and ageusia are common symptoms in SARS-CoV-2 infection and should be considered when guiding screening and in the recommendations of self-isolation. HCW with direct patient contact are at increased risk of SARS-CoV-2 infection, and continued efforts, such as a wider implementation of RT-PCR screening in both HCW and patients as well as adequate PPE, are warranted to protect HCW and to reduce transmission from HCW to patients and to the community. These measures may limit the ongoing pandemic of SARS-CoV-2.

## Methods

**Hospital setting**. This cross-sectional study was conducted at Danderyd Hospital, an acute-care 530-bed hospital providing both general and specialized hospital care. With a catchment area of 600,000 individuals, a total of 722 patients with confirmed COVID-19 had been admitted to the hospital during the study period. The hospital guidelines required personal protective equipment (PPE) to be worn by all employees with direct contact with confirmed or suspected SARS-CoV-2 infected patients, but not with direct contact with patients not suspected to be infected with SARS-CoV-2. Face shields, FFP3/FFP2/N95 masks and sleeveless plastic aprons were recommended in general wards during aerosol generating procedures (AGPs), and at all times in the ICU. Aerosol filtering face masks were replaced by surgical face masks IIR during non-AGP. Due to PPE shortage, face shields were reused after cleaning and disinfection, both by one individual for multiple encounters and between individuals. During the study period, reverse-transcriptase polymerase chain reaction (RT-PCR) viral RNA detection of nasopharyngeal or oropharyngeal swabs was not available for hospital employees, regardless of symptoms. No symptom screening was performed, but hospital employees were encouraged to stay at home if they had respiratory symptoms or fever, and to remain at home for 2 days post symptoms. Patients presenting with COVID-19 symptoms (defined as cough, fever, dyspnea, rhinitis, sore throat, headache, nausea or myalgia) were tested with RT-PCR, but patients without these symptoms were not tested.

**Study population**. All employees at Danderyd Hospital ($n = 4375$) were invited by e-mail and/or on the hospital intranet to participate in the study. Consecutive study inclusion took place between 15 April and 8 May, 2020. Participants were eligible to participate in the study irrespective of whether they had had symptoms since the COVID-19 outbreak onset or not. There were no exclusion criteria. Informed consent, study inclusion and appointment for blood sampling were obtained using a smartphone-based app and verification through electronic identification and signature. Participants completed a questionnaire comprising demographics (age and sex), self-reported predefined prior symptoms compatible with COVID-19 since 1 January 2020, occupation, work location and self-reported exposure to patients infected with COVID-19 prior to blood sampling. The questionnaire consisted of 11 predefined symptoms (fever, headache, anosmia, ageusia, cough, malaise, common cold, abdominal pain, sore throat, shortness of breath) including the alternative "no ongoing or prior symptoms since 1 January 2020". Fever could be either subjective or verified by temperature as no specifications regarding this was required. Participants were asked to grade the symptoms as mild or severe, and to document symptom onset. Participants were asked to state their occupation as physician, nurse, assisting nurse, or other healthcare personnel (comprising occupations such as physiotherapists, speech therapists, and occupational therapists), as well as department of employment in free text. Due to frequent relocation of hospital staff during the pandemic, participants were further asked to state whether they had direct patient contact and whether they had worked in COVID-19 departments, (COVID-19 zone of the emergency department, COVID-19 transit ward (admitting patients awaiting laboratory confirmation of COVID-19 infection), COVID-19 general wards and COVID-19 intermediate and intensive care units). The portion of study participants answering all questions was 96.3%.

**Serological analyses of antibodies**. Plasma samples were prepared from whole blood following centrifugation for 20 min at 2000 $g$ at room temperature, viral inactivation for 30 min at 56 °C, and stored at −80 °C until further analyses. Serological analyses of IgG were performed. IgG reactivity was measured towards three different virus protein variants (Spike trimers comprising the prefusion-stabilized spike glycoprotein ectodomain[35] (in-house produced, expressed in HEK and purified using a C-terminal Strep II tag) Spike S1 domain (Sino Biological, expressed in HEK and purified using a C-terminal His-tag), and Nucleocapsid protein (Acro Biosystems, expressed in HEK and purified using a C-terminal His-tag), and analyzed using a multiplex antigen bead array in high throughput 384-plates format using a FlexMap3D (Luminex Corp)[36]. Plasma IgG was captured on the antigen coated beads and detected by fluorescent goat anti-hIgG (Invitrogen, H10104). The read out consisted of the bead-based median fluorescent intensity (MFI) and count of number of beads for each antigen (bead ID) in each sample. To be assigned as an IgG positive sample, reactivity against at least two of the three different variants of the viral antigens was required and the cutoff was defined as signals above the mean+6 SD of the 12 negative controls included in each analysis. These cutoff values were also used when analyzing 243 positive control samples (defined as PCR-positive individuals with mild to severe symptoms sampled more than 16 days after onset) and 442 negative control samples (defined as samples collected 2019 and earlier, including confirmed infections with non-SARS-CoV-2 corona viruses). Based on these control samples, the method was calculated to have 99.2% sensitivity (99.6%, 99.2%, 96.7%, respectively, for the three antigens individually) and 99.8% specificity (98.9%, 99.1%, 98.4%, respectively, for the three antigens individually), see Supplementary Table 1 and Supplementary Fig. 1. Four positive controls were re-run on every assay-plate and had a mean inter-assay coefficient of variation of 10.1% (8.0–13.3%) based on absolute intensity levels.

**Statistical analyses**. Descriptive analyses were made on baseline characteristics and the number of observations, presented as numbers and percentages. Categorical variables are presented as proportions, compared with the Fisher's exact test and reported as odds ratio (OR) and 95% confidence interval (CI). Continuous variables are presented as means with standard deviations (SD) and compared with the unpaired $t$-test. The group of HCW with patient-related work ($n = 1764$) was compared to the group of HCW with non-patient-related work ($n = 305$). The group of HCW with patient-related work was further divided into COVID-19 patient contact and non-COVID-19 patient contact, and these two sub groups were compared to each other. Within the group of HCW with patient-related work, occupations as assisting nurses ($n = 428$), nurses ($n = 636$), other healthcare personnel ($n = 254$), and physicians ($n = 439$) were compared to the group with non-patient-related work. Statistical analyses and visualizations were performed in R, using packages tidyverse, lubridate, rlang, pander, knitr, and UpSetR (RStudio Team 2019, Boston, USA).

The study complied with the declaration of Helsinki, and informed consent was obtained by all participants. The study protocol was approved by the Stockholm Ethical Review Board (dnr 2020-01653). This report conforms to the STROBE guidelines.

## Data availability

The anonymized datasets generated during and/or analyzed during the current study are available from the corresponding author on reasonable request. Source data are provided with this paper.

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

## Acknowledgements

This study was funded by Region Stockholm, Knut and Alice Wallenberg foundation, Science for Life Laboratory (SciLifeLab), Erling-Persson family foundation, the Swedish Research Council, Christian and Jennifer Dahlberg, Atlas Copco, and EU grant (CoroNab). The authors are grateful to Carola Jonsson, Sofie Lundin, Tina Johnsson, Camilla Redhevon, Mitra Samadi, Bibi Fundell, Ami Sjöblom, Sarah Juhlin, Emelie Karlsson, Lina Petersson, Julia Alm Baker, Eva Edström, Frehiywet Tes, Anna Weimer and Mery Millberg at Danderyd Hospital for assisting in administration and blood sampling. We thank Richard Scholvin for technical support with the smartphone app and assisting with data information, and Carina Rudberg and Christina Einarsson for assisting with data collection. We are grateful to David Just, Cornelia Westerberg, Mimmi Olofsson and Sara Mravinacova for technical assistance in performing and developing the serology assays. We are grateful to Karolinska University Laboratory and the Public Health Agency of Sweden for providing negative and positive control samples. We also would like to thank Gunilla Karlsson Hedestam for fruitful discussions.

## Author contributions

C.T., A.S.R., S.e.H., M.P., S.o.H., and P.N. designed the study, collected data, performed analyses, interpreted data and wrote the manuscript. A.M., A.J.F., K.A., H.N., L.G., A.-C.S., J.O., E.A., C.H., Sh.B., S.o.B., E.P., R.S., H.T., and M.H. performed analyses, statistical analyses, interpreted data and commented on the manuscript. L.H., B.M., and G.M.I. contributed to the development of the serological assay and commented on the manuscript. All authors have approved the final version of the manuscript.

## Funding

## Competing Interests

The authors declare no competing interests.
