## [Peer Review File · Nature Communications]

REVIEWER COMMENTS

Reviewer #1 (Remarks to the Author):

In the study "SARS-CoV-2 exposure, symptoms and seroprevalence in health care workers" the authors investigate the seroprevalence in a large Swedish hospital among HCW and non-HCW personal and correlate these results with self-reported symptoms. Overall the study is well written and the conclusions are largely supported by the data presented. Also the limitations are clearly stated. However, before the study can be accepted for publication there are a couple of minor points that need to be revised:

Minor:

Page 4 first line: It spells coronavirus, not "corona virus"

Page 6: When asked for fever as a symptom, was this subjective fever, i.e. the patient felt feverish but did not measure his temperature or objective fever, i.e. the patient actually measured his temperature.

Page 7: was there any difference in sensitivity between mild and severe cases in the validation data set

Page 7 and 8: Concerning the determination of the cut-off for their assay the authors state on page 7 "The cutoff values used in the study was chosen after analyzing 154 positive samples (defined as PCR-positive individuals with mild to severe symptoms sampled more than 14 days after onset) and 321 negative samples" and on page 8 "cutoff was defined as signals above the mean+6 SD of the 12negative controls included in each analysis". Please clarify which method was used to calculate the cut-off.

Page 8: In my opinion the validation data is an important part of your study, as it is the basis for the reliability of the results of the whole paper. Therefore, I suggest to include the mfi data of the 154 COVID patients and 321 neg. ctrls. as a scatter plot in the supplement and also a table with the number of positive/neg. for each antigen and the calculation of specificity and sensitivity (including 95% CI).

Table 1: The setup of table 1 is a bit confusing to me, as e.g. the "missing" data is reported for every occupation. I.e. it looks like for the physicians there are 80 persons were the exposure is missing, which contradicts itself. I would suggest to include the missing data in another row (instead of a column) below the "direct contact with covid patients" row.

Page 10: How do the authors define mild and severe symptoms? Is this according to the WHO classification of mild, severe and critical cases? Please clarify.

Results in general: I think it would be very helpful to include a table with the % seroprevalence (+95%CI) listed by occupation, sex and exposure risks.

Discussion in general: While the discussion is clearly written, the authors fail to cite a study by Xu et al (1) that already described an occupational risk for HCW compared to the general population. This also make the claim to be the "the first large serological investigation supporting an occupational risk of SARS-CoV-2 transmission to HCW" unsubstantiated. Interestingly, Xu et al. only found a seroprevalence of 3.8% in HCW in the severely affected city of Wuhan which is a substantial discrepancy to the results of this study. I think, it is very important to discuss the result of both studies and discuss why the seroprevalence in HCW in Sweden is so much higher than in Wuhan.

1. Xu, X. et al. Seroprevalence of immunoglobulin M and G antibodies against SARS-CoV-2 in China. Nat Med (2020) doi:10.1038/s41591-020-0949-6.

Reviewer #2 (Remarks to the Author):

Overall:

This is an interesting manuscript demonstrating high SARS-CoV-2 seroprevalance across

healthcare personnel in one hospital, during a time of ongoing transmission in Sweden. However other manuscripts have already described seroprevalence investigations in healthcare personnel, therefore these findings are not particularly novel, and other seroprevalence investigations in healthcare personnel (HCP) should be referenced here e.g. <https://doi.org/10.1093/cid/ciaa936> , <https://doi.org/10.1093/cid/ciaa761> . There is also the concern that infection prevention and control precautions are not discussed in the context of such a high seroprevalence in healthcare personnel. In particular, why are so many HCP are seropositive despite PPE protocols, and whether these HCP may have been able to transmit SARS-CoV-2 to patients and other HCP in health care settings. What actions were taken to protect HCP from infection, and to prevent symptomatic HCP from attending work given these findings? Additionally, the method of data analysis is not clear - there are no specifics as to why the three out of four symptoms (anosmia, ageusia, fever and malaise) was of such great focus, when other combinations could have been investigated.

General Comments:

A few key questions exist around the sampled population included in this study. First, it is mentioned that the only exclusion criteria was absence from work due to illness. It is not clear from this statement whether this criterion included any single day of absence from work since January 2020, absence the time of study (15th April – 8th May), or absence from work throughout the entire potential exposure period i.e. since January 1, 2020. Given that absence from work resulted in participant exclusion, this brings up the concern that the investigation will underestimate seroprevalence, as individuals who were absent from work due to symptomatic COVID-19 infection would be excluded. More concerning, is that if all individuals who were absent from work were excluded from this study, then it infers that study participants who had symptoms attended work during their COVID-19 illness, with many reporting severe symptoms. It should have been that case that some sort of HCP symptom screening should have been in place by April – May 2020 to prevent symptomatic HCP from working, but this is not discussed.

Odds ratios should be presented along with 95% confidence intervals.

It is not necessary to reflect exceedingly small p values, would reflect <0.01 , rather than such small numbers as described in this manuscript.

Abstract

It would be helpful to have in the abstract that the seroprevalence in this study was significantly higher than the national seroprevalence during the same time period.

Methods

It is concerning that rRT-PCR testing “was not available for hospital employees” – is it just that any results of such testing were not available for this manuscript? If testing was not available at all for hospital employees who continued to work, with symptoms of SARS-CoV-2 at the time, this reflects a substantial breakdown of ideal infection control practices in the hospital and should be discussed as to why this was the case.

How was ‘patient contact’ defined? Direct physical contact with the patient? Presence in the same room for >10 minutes within < 6 feet? Does contact with infected secretions from a patient count?

Results

It would be interesting to know what the breakdown of ‘other healthcare personnel’ is, if this is available, particularly among those with and without patient contact. Are these office workers, receptionists, environmental service workers? Environmental services workers may spend a long time around patients in their room but may not be considered to have direct patient contact,

depending on the definition used.

Table 1. 'Missing' categories would be more easily understood as a 'row' for each respective categorical variable rather than a column, e.g. male, female, missing.

Fig 1. Would be helpful to have the total number of seropositive/seronegative participants mentioned within the appropriate figure or legend. It is not clear why axes are labeled intersection size and set size when in the legend it says these axes reflect number of participants. Additionally the numbers are not needed above the bars when they can be inferred from the axis – perhaps %s may be more helpful here so the two graphs are directly comparable.

The combination of 'three out of the four following symptoms: anosmia, ageusia, fever and malaise' is mentioned as having high PPV and NPV, however it is not specified why this combination was chosen. For example, the PPV and NPV of "either of anosmia or ageusia" is not considered, which has considerably higher sensitivity.

'Three out of the four following symptoms: anosmia, ageusia, fever and malaise': Sensitivity 47%, Specificity 96%, PPV 75%, NPV 88%

'Any anosmia or ageusia': Sensitivity 58%, Specificity 94%, PPV 70%, NPV 91%

I would suggest a more thorough investigation into the best symptom sets for highest sens/spec/PPV/NPV – or a specific explanation for why this 3 out of 4 criteria was chosen, especially above an 'anosmia/ageusia' approach which the data indicates may be more appropriate.

Discussion

The implications of such a high proportion of HCP seropositive for SARS-CoV-2, particularly with the implication that none of them were absent from work due to their illnesses, and that no rRT-PCR testing was available to HCP, is that HCP could easily spread SARS-CoV-2 throughout the hospital, both to patients and colleagues. The implications of this, nor the HCP screening and other infection control precautions (e.g. encouraging workers to remain home if they have any symptoms), are not described in this article, which is a cause for concern. It is likely that many of these measures were in place and they should be discussed in this manuscript.

There should be more discussion around why HCP with patient contact, but without known contact with COVID-19 patients, might have higher seroprevalence than those without patient contact, rather than just saying this is unclear. You hint that it may be difficult to distinguish SARS-CoV-2 positive patients from SARS-CoV-2 negative patients, but there should be further discussion. What is the hospital doing to identify COVID-19 patients and has any of this changed? This is particularly important given the knowledge of asymptomatic transmission. Are all inpatients tested for SARS-CoV-2 on admission since this investigation? What are the criteria for screening for patients for COVID-19?

It is also not clear that selection bias would apply equally to all subgroups – if some subgroups are more likely to be ill and off work for example, this would increase selection bias disproportionately among this group.

Response to referee comments,

We would like to thank the reviewers for their critical and helpful comments, which we have earnestly worked to address and which we believe greatly strengthen the conclusions and impact of our study. A point-by-point response to each comment is below, and major changes to the text have been noted in blue.

Referee 1:

Reviewer #1 (Remarks to the Author):

In the study “SARS-CoV-2 exposure, symptoms and seroprevalence in health care workers” the authors investigate the seroprevalence in a large Swedish hospital among HCW and non-HCW personal and correlate these results with self-reported symptoms. Overall the study is well written and the conclusions are largely supported by the data presented. Also the limitations are clearly stated. However, before the study can be accepted for publication there are a couple of minor points that need to be revised:

Minor:

1. Page 4 first line: It spells coronavirus, not “corona virus”

Response: Thank you, we have corrected this error.

2. Page 6: When asked for fever as a symptom, was this subjective fever, i.e. the patient felt feverish but did not measure his temperature or objective fever, i.e. the patient actually measured his temperature.

Response: It was not specified whether fever was subjective or verified by temperature in the questionnaire. We have now clarified this in methods page 6.

3. Page 7: was there any difference in sensitivity between mild and severe cases in the validation data set

Response: Unfortunately, we do not have access to information regarding the severity of symptoms for the positive controls.

4. Page 7 and 8: Concerning the determination of the cut-off for their assay the authors state on page 7 “The cutoff values used in the study was chosen after analyzing 154 positive samples (defined as PCR-positive individuals with mild to severe symptoms sampled more than 14 days after onset) and 321 negative

samples” and on page 8 “cutoff was defined as signals above the mean+6 SD of the 12negative controls included in each analysis”. Please clarify which method was used to calculate the cut-off.

Response: We are grateful for this comment and agree that the previous description needed clarification. The text has now been rephrased accordingly, methods, page 8.

5. Page 8: In my opinion the validation data is an important part of your study, as it is the basis for the reliability of the results of the whole paper. Therefore, I suggest to include the mfi data of the 154 COVID patients and 321 neg. ctrls. as a scatter plot in the supplement and also a table with the number of positive/neg. for each antigen and the calculation of specificity and sensitivity (including 95% CI).

Response: We agree that this is a very important component. We have now added a supplementary table with details on the number of positive and negative controls and their reactivities, as well as the specificity and sensitivity for each of the antigens, including 95% CI. We have also added, as suggested, a supplementary figure where the intensity for each of the controls for the different antigens are visualized. It should be noted that we have been able to increase the number of both negative (now 442) and positive (now 243) controls since the first submission and the text in the Serological analyses of antibodies section has been updated accordingly.

6. Table 1: The setup of table 1 is a bit confusing to me, as e.g. the “missing” data is reported for every occupation. I.e. it looks like for the physicians there are 80 persons were the exposure is missing, which contradicts itself. I would suggest to include the missing data in another row (instead of a column) below the “direct contact with covid patients” row.

Response: We thank the referee for this relevant comment, and agree that the presentation of missing data is not clear. Missing data is now moved to rows below relevant variables for clarity in Table 1 and the new Table 2 (see below).

7. Page 10: How do the authors define mild and severe symptoms? Is this according to the WHO classification of mild, severe and critical cases? Please clarify.

Response: The symptom gradings as “mild” or “severe” were not specified further in the questionnaire, as we attempted to create the questionnaire as simple as possible to obtain a high response rate. We acknowledge that these gradings are subjective, which is a limitation. We have now added a sentence discussing this limitation in discussion page 21.

8. Results in general: I think it would be very helpful to include a table with the % seroprevalence (+95%CI) listed by occupation , sex and exposure risks.

Response: We agree that this data should be clearly visible, and have now added this in a new Table 2.

9. Discussion in general: While the discussion is clearly written, the authors fail to cite a study by Xu et al (1) that already described an occupational risk for HCW compared to the general population. This also make the claim to be the “the first large serological investigation supporting an occupational risk of SARS-CoV-2 transmission to HCW” unsubstantiated. Interestingly, Xu et al. only found a seroprevalence of 3.8% in HCW in the severely affected city of Wuhan which is a substantial discrepancy to the results of this study. I think, it is very important to discuss the result of both studies and discuss why the seroprevalence in HCW in Sweden is so much higher than in Wuhan.

1. Xu, X. et al. Seroprevalence of immunoglobulin M and G antibodies against SARS-CoV-2 in China. Nat Med (2020) doi:10.1038/s41591-020-0949-6.

Response: We thank the referee for these valuable comments.

We have now added relevant references of recently published seroprevalence investigations in healthcare workers (Xu et al, 2020, Garcia-Basteiro et al, 2020 and Steensels et al, 2020), and acknowledge the relatively high seroprevalence found in our study. Although these discrepancies could result from different study design and serological assays, infection control and prevention strategies may (as pointed out by both referees below) also play a role. There were several differences in infection prevention and control precautions between our hospital and, for example, the Belgian hospital (Steensels et al, 2020) in which only 6.4% of 3056 healthcare workers were seropositive. Notably, this study was similar to ours in study design, time period of study inclusion and sample size. The main differences were: 1) RT-PCR testing was not available for HCW in our hospital (regardless of symptoms), 2) RT-PCR testing was limited to in-hospital patients with typical covid-19 symptoms (as opposed to testing all in-house patients), 3) PPE was limited to covid-19 patient contact, and 4) aerosol filtering face masks were limited to aerosol generating procedures. These discrepancies may have contributed to the relatively high seroprevalence found in our study, and are now discussed in detail, (discussion, page 18-20).

We furthermore clarify the analyses of symptoms associated with SARS-CoV-2 (see referee comment nr 20). We have changed the sentence “the first large serological investigation supporting an occupational risk of SARS-CoV-2 transmission to HCW” in discussion, page 16, to “This large serological investigation supports an occupational risk of SARS-CoV-2 transmission to HCW, exceeding the risk presented in prior studies (refs 8, 9, 13)”.

Reviewer #2 (Remarks to the Author):

Overall:

10. This is an interesting manuscript demonstrating high SARS-CoV-2 seroprevalance across healthcare personnel in one hospital, during a time of ongoing transmission in Sweden. However other manuscripts have already described seroprevalence investigations in healthcare personnel, therefore these findings are not particularly novel, and other seroprevalence investigations in healthcare personnel (HCP) should be referenced here e.g. <https://doi.org/10.1093/cid/ciaa936> , <https://doi.org/10.1093/cid/ciaa761> . There is also the concern that infection prevention and control precautions are not discussed in the context of such a high seroprevalence in healthcare personnel. In particular, why are so many HCP are seropositive despite PPE protocols, and whether these HCP may have been able to transmit SARS-CoV-2 to patients and other HCP in health care settings. What actions were taken to protect HCP from infection, and to prevent symptomatic HCP from attending work given these findings?

Additionally, the method of data analysis is not clear - there are no specifics as to why the three out of four symptoms (anosmia, ageusia, fever and malaise) was of such great focus, when other combinations could have been investigated.

Response: In line with the response to Reviewer 1 (comment 9), we acknowledge that further discussion of recently published data is relevant, and have added this to the manuscript. In addition, we agree with the reviewer that the discussion on possible inadequacies in infection prevention and control precautions underlying the relatively high seroprevalence found among health care workers in our study is important, and have now included this in the discussion part of the manuscript (see below). We furthermore clarify the analyses of symptoms associated with SARS-CoV-2 (see below).

General Comments:

11. A few key questions exist around the sampled population included in this study. First, it is mentioned that the only exclusion criteria was absence from work due to illness. It is not clear from this statement whether this criterion included any single day of absence from work since January 2020, absence the time of study (15th April – 8th May), or absence from work throughout the entire potential exposure period i.e. since January 1, 2020. Given that absence from work resulted in participant exclusion, this brings up the concern that the investigation will underestimate seroprevalence, as individuals who were absent from work due to symptomatic COVID-19 infection would be excluded. More concerningly, is that if all individuals who were absent from work were excluded from this study, then it infers that study participants who had symptoms attended work during their COVID-19 illness, with many reporting severe symptoms. It should have been that case that some sort of HCP symptom screening should have been in place by April – May 2020 to prevent symptomatic HCP from working, but this is not discussed.

Response: We acknowledge that the inclusion and exclusion criteria regarding absence from work is not clearly described, and we have now clarified this in methods, page 6. The sentence “The only exclusion criterion was absence from work due to illness” is now removed, and replaced by “There were no exclusion criteria”.

All employees were eligible for inclusion in the study, regardless of whether they had been absent from work due to illness. However, employees absent from work due to illness *during the entire study period* (15th April – 8th May), were not able to participate in the study since they were not at work. As implied by the reviewer, absence from work was therefore not an *exclusion criterion*, but rather a possible cause of *selection bias*. Since the study period spanned over three weeks, we anticipate that the group of employees not able to participate due to illness was small. Although we assume that this selection bias is small, this potential limitation is now added in discussion, page 21.

Participants were asked to state *prior symptoms*, this is now clarified in methods, page 6. Since participants were included while at work, we did not expect a high prevalence of ongoing symptoms.

Notably, there was no RT-PCR screening of HCW in place at the time of the study, regardless of symptoms. HCW screening was implemented first mid-June. The reason for this was a limited testing availability. This may well have contributed substantially to the high seroprevalence among our participants, as well as the spread of SARS-CoV-2 throughout the hospital, both to patients and colleagues. As we acknowledge that this is of high concern, we have now added a section discussing this on page 19-20.

12. Odds ratios should be presented along with 95% confidence intervals.

Response: We thank the reviewer for this suggestion. 95% confidence intervals for OR are now presented throughout the manuscript.

13. It is not necessary to reflect exceedingly small p values, would reflect <0.01, rather than such small numbers as described in this manuscript.

Response: P-values are now replaced by 95% confidence intervals.

Abstract

14. It would be helpful to have in the abstract that the seroprevalence in this study was significantly higher than the national seroprevalence during the same time period.

Response: We agree with this suggestion and we have added this sentence in the abstract (the seroprevalence was reported for Stockholm, so we refer to the regional seroprevalence).

Methods

15. It is concerning that rRT-PCR testing “was not available for hospital employees” – is it just that any results of such testing were not available for this manuscript? If testing was not available at all for hospital employees who continued to work, with symptoms of SARS-CoV-2 at the time, this reflects a substantial breakdown of ideal infection control practices in the hospital and should be discussed as to why this was the case.

Response: As stated above, RT-PCR screening of HCW was not in place at the time of the study, regardless of symptoms. HCW were informed to stay at home if they had symptoms. We agree that this reflects a substantial breakdown of ideal infection control, along with other important inadequacies, such as the policy of limiting RT-PCR testing to in-hospital patients with typical covid-19 symptoms, the limitation of PPE to direct contact with confirmed or suspected SARS-CoV-2 infected patients, and the re-use of face shields (see below). These issues have now been added to methods and discussion. Importantly, the inadequate PPE and lack of RT-PCR testing was hampered by a limited supply and testing availability, and was likely similar in other hospitals and health care settings around the nation.

16. How was ‘patient contact’ defined? Direct physical contact with the patient? Presence in the same room for >10 minutes within < 6 feet? Does contact with infected secretions from a patient count?

Response: Patient contact was defined by the self-reporting of “direct patient contact” by the participants. We believe that this classification is superior to occupational location in the hospital data base since a large portion of study participants were temporarily re-located during the covid-19 pandemic, as discussed on page 19. However, the type of contact, such as time in the same room, was not specified. We acknowledge this limitation, and a sentence discussing this is now added on page 21.

Results

17. It would be interesting to know what the breakdown of ‘other healthcare personnel’ is, if this is available, particularly among those with and without patient contact. Are these office workers, receptionists, environmental service workers? Environmental services workers may spend a long time around patients in their room but may not be considered to have direct patient contact, depending on the definition used.

Response: As stated above (comment nr 16), patient contact was defined by the self-reporting of direct patient contact by the participants, regardless of occupation. The group “other health care staff” in the occupation groupings comprised occupations such as physiotherapists, speech therapists, and occupational therapists, but we do not have the breakdown of these occupations. Sanitation and cleaning personnel are not employed by Danderyd Hospital, and thus not eligible for inclusion in the study.

18. Table 1. ‘Missing’ categories would be more easily understood as a ‘row’ for each respective categorical variable rather than a column, e.g. male, female, missing.

Response: As stated above (comment nr 6), we agree that the presentation of missing data is not clear. Missing data is now moved to rows below relevant variables for clarity in Table 1 and new Table 2.

19. Fig 1. Would be helpful to have the total number of seropositive/seronegative participants mentioned within the appropriate figure or legend. It is not clear why axes are labeled intersection size and set size when in the legend it says these axes reflect number of participants. Additionally the numbers are not needed above the bars when they can be inferred from the axis – perhaps %s may be more helpful here so the two graphs are directly comparable.

Response: We thank the referee for this comment, and we have now stated the total number of participants for each figure. Axes are now labeled as coprevalence, which we agree is more correct. We furthermore replaced the numbers with percent above the bars which we agree is more informative.

20. The combination of ‘three out of the four following symptoms: anosmia, ageusia, fever and malaise’ is mentioned as having high PPV and NPV, however it is not specified why this combination was chosen. For example, the PPV and NPV of “either of anosmia or ageusia” is not considered, which has considerably higher sensitivity.

‘Three out of the four following symptoms: anosmia, ageusia, fever and malaise’: Sensitivity 47%, Specificity 96%, PPV 75%, NPV 88%
‘Any anosmia or ageusia’: Sensitivity 58%, Specificity 94%, PPV 70%, NPV 91%

I would suggest a more thorough investigation into the best symptom sets for highest sens/spec/PPV/NPV – or a specific explanation for why this 3 out of 4 criteria was chosen, especially above an ‘anosmia/ageusia’ approach which the data indicates may be more appropriate.

Response: This is an excellent point and we acknowledge that the choice of symptom sets is not clear.

The first symptom set, anosmia *and/or* ageusia, malaise and fever was chosen since these were the symptoms found to have the highest OR in our data set. The second symptom set, anosmia *and/or* ageusia, malaise and cough was chosen in light of the recently published population based real-time tracking of self-reported symptoms predicting covid-19 (Menni et al, 2020). This large study found, in line with our data, a strong association between anosmia/ageusia and a positive test result, and that the combination of anosmia, ageusia, malaise, persistent cough and loss of appetite resulted in the best symptom combination model to predict probable infection. We therefore aimed to test this symptom combination in our data set. Since we did not have information on “loss of appetite”, we created the symptom set anosmia *and/or* ageusia, malaise and cough. We have now clarified these choices in results, page 12.

We also acknowledge the potential strength in the combination anosmia *and/or* ageusia alone, and we have done further analyses of this symptom set revealing, as the referee correctly points out, the strongest predictive value among the combinations. We thank the referee for this suggestion, and have now added this symptom set in results, page 13.

Discussion

21. The implications of such a high proportion of HCP seropositive for SARS-CoV-2, particularly with the implication that none of them were absent from work due to their illnesses, and that no rRT-PCR testing was available to HCP, is that HCP could easily spread SARS-CoV-2 throughout the hospital, both to patients and colleagues. The implications of this, nor the HCP screening and other infection control precautions (e.g. encouraging workers to remain home if they have any symptoms), are not described in this article, which is a cause for concern. It is likely that many of these measures were in place and they should be discussed in this manuscript.

Response: We agree with the referee.

As stated above (comment nr 11), absence from work was not an exclusion criterion, which has now been removed from the methods section.

RT-PCR testing was not available for HCW (see above). We have added this in methods and discuss possible implications of this, and other limitations in infection control preventions, in discussion, pages 18-20.

22. There should be more discussion around why HCP with patient contact, but without known contact with COVID-19 patients, might have higher seroprevalence than those without patient contact, rather than just saying this is unclear. You hint that it may be difficult to distinguish SARS-CoV-2 positive patients from SARS-CoV-2 negative patients, but there should be further discussion. What is the hospital doing to identify COVID-19 patients and has any of this changed? This is particularly important given the knowledge of asymptomatic transmission. Are all inpatients tested for SARS-CoV-2 on

admission since this investigation? What are the criteria for screening for patients for COVID-19?

Response: We agree that the high seroprevalence among participants with non-covid patient contact is disturbing. The policy during the study period was to limit RT-PCR testing to in-hospital patients with typical covid-19 symptoms (as opposed to testing all in-house patients), see referee comment nr 15. The admission of SARS-CoV-2 patients to non-covid wards may not only have contributed to a high infection rate among HCW in these non-covid wards, but also to transmission from HCW to patients and between patients. Notably, the policy during the study period was to limit PPE to covid-19 patient contact (see referee comment nr 15), and HCW in non-covid wards did not wear PPE. These policies, along with above mentioned shortcomings in prevention measures, are now discussed on pages 18-20 as potential contributors to the high seroprevalence.

23. It is also not clear that selection bias would apply equally to all subgroups – if some subgroups are more likely to be ill and off work for example, this would increase selection bias disproportionately among this group.

Response: We acknowledge that some work groups may have been more likely to be ill and stay home from work, but since this was not an exclusion criteria (which has now been clarified, see above) we do not think this contributed to a significant selection bias disproportionately among this groups.

We thank the reviewers for the responses. We believe that the above comments were relevant and valuable, and that our changes in response have improved the manuscript. We hope that you will find the revised manuscript acceptable for publication in Nature Communications.

Yours sincerely,

Charlotte Thålin, M.D., Ph.D. (Senior and Corresponding author)
Karolinska Institutet, Danderyd Hospital
18288 Stockholm, Sweden
charlotte.thalin@sll.se, Phone: +46709565120

REVIEWER COMMENTS

Reviewer #2 (Remarks to the Author):

The edits made have greatly improved the clarity of the manuscript, and the resulting conclusions made. There are a few remaining issues which should ideally be addressed below.

Introduction

I would still encourage the authors to conduct a literature review on previous serologic studies in health care workers as this manuscript does not consider the breadth of prior studies on this topic. It would be helpful to quote the range of seroprevalence seen in health care workers in previous studies here. Note there are more studies looking at healthcare worker seroprevalence beyond the two references quoted e.g. <https://doi.org/10.1016/j.jcv.2020.104437> , <https://doi.org/10.1093/cid/ciaa936>, <https://doi.org/10.1101/2020.06.24.20135038> , and more.

Methods

It is stated that hospital employees were encouraged to stay at home if symptomatic (presumably only while symptomatic, not for any defined period of time), but was any symptom screening performed?

Also while it is mentioned that patients with symptoms were tested for COVID-19, it should be mentioned that healthcare workers with symptoms were not tested, and what procedures were implemented instead e.g. told to isolate at home until no longer symptomatic.

Could you define what 'typical' symptoms led patients to be tested for COVID-19. This will help understand which patients may have been missed.

Discussion

It is mentioned that RT-PCR testing of HCP was not performed, but this alone would not make any difference – it is testing and subsequent isolation of infected HCP to prevent spread that could make a difference, so would suggest adjusting discussion accordingly.

The newly added paragraph is helpful, and contains useful discussion of the lack of infection control practices compared to those documented in other studies with lower prevalence . However the paragraph itself is very long and somewhat confusing to follow, suggest editing for clarity, and emphasizing key points in the final conclusions e.g. direct patient contact to non covid patients was a risk factor - likely as this was combined with HCW not wearing appropriate PPE and a lack of screening of these patients etc.

A limitation of this study was that exposures to COVID-19 positive individuals outside of the hospital were not documented.

Defining 'typical' symptoms in the methods will help understand the discussion on which hospitalized patients would not have been tested here.

Response to referee comments,

We would like to thank the reviewer for the additional comments, to which we believe our responses have improved the manuscript in this second revision. Below is a point-by-point reply to each comment, and major changes to the text have been noted in blue.

Reviewer #2 (Remarks to the Author):

The edits made have greatly improved the clarity of the manuscript, and the resulting conclusions made. There are a few remaining issues which should ideally be addressed below.

1. Introduction

I would still encourage the authors to conduct a literature review on previous serologic studies in health care workers as this manuscript does not consider the breadth of prior studies on this topic. It would be helpful to quote the range of seroprevalence seen in health care workers in previous studies here. Note there are more studies looking at healthcare worker seroprevalence beyond the two references quoted e.g. <https://doi.org/10.1016/j.jcv.2020.104437> , <https://doi.org/10.1093/cid/ciaa936>, <https://doi.org/10.1101/2020.06.24.20135038>, and more.

Response: We thank the reviewer for this comment, and we have now conducted a new literature search of recently published serological studies in health care settings. As suggested, we now refer to the above new studies, as well as a recent Danish report ([https://doi.org/10.1016/S1473-3099\(20\)30589-2](https://doi.org/10.1016/S1473-3099(20)30589-2)). We quote the range of seroprevalence seen in these seven studies, although they are all significantly lower than the seroprevalence found in this study.

2. Methods

It is stated that hospital employees were encouraged to stay at home if symptomatic (presumably only while symptomatic, not for any defined period of time), but was any symptom screening performed?

Response: We acknowledge that further clarification is warranted. Hospital workers were encouraged to stay at home if they had respiratory symptoms or fever, and to remain home for 2 days post symptoms. No symptom screening was performed. We have now clarified this in methods, page 6.

3. Also while it is mentioned that patients with symptoms were tested for COVID-19, it should be mentioned that healthcare workers with symptoms were

not tested, and what procedures were implemented instead e.g. told to isolate at home until no longer symptomatic.

Response: We have now clarified that hospital workers were not tested for covid-19, *regardless of symptoms*, in methods, page 6. As stated above, and now clarified in methods, page 6, hospital workers with respiratory symptoms or fever were encouraged to stay home, and to remain home for 2 days post symptoms.

4. Could you define what ‘typical’ symptoms led patients to be tested for COVID-19. This will help understand which patients may have been missed.

Response: We agree that this needs to be specified, and we have now defined the symptoms which led patients to be tested (cough, fever, dyspnoea, rhinitis, sore throat, headache, nausea or myalgia) in methods, page 6.

5. Discussion

It is mentioned that RT-PCR testing of HCP was not performed, but this alone would not make any difference – it is testing and subsequent isolation of infected HCP to prevent spread that could make a difference, so would suggest adjusting discussion accordingly.

Response: We thank the reviewer for this remark, and we have adjusted the sentence to “RT-PCR testing and subsequent isolation of infected HCW”, page 19.

6. The newly added paragraph is helpful, and contains useful discussion of the lack of infection control practices compared to those documented in other studies with lower prevalence . However the paragraph itself is very long and somewhat confusing to follow, suggest editing for clarity, and emphasizing key points in the final conclusions e.g. direct patient contact to non covid patients was a risk factor - likely as this was combined with HCW not wearing appropriate PPE and a lack of screening of these patients etc.

Response: We agree with the reviewer that this paragraph was too long, at times repetitive, and confusing to follow. We have now shortened it, and edited for clarity. Key points are that the following practices may have contributed to nosocomial spread: 1) lack of RT-PCR testing and subsequent isolation of infected HCW, 2) lack of RT-PCR testing of all in-hospital patients (regardless of typical covid-19 symptoms which are now defined in methods), 3) lack of PPE in contact with patients in non-covid wards, 4) restriction of aerosol filtering face masks to AGP, and 5) the reuse of manually cleaned and disinfected face shields. These key points are now mentioned only once, and in this order, page 20.

7. A limitation of this study was that exposures to COVID-19 positive individuals outside of the hospital were not documented.